# Characterization of the Metabolome of Breast Tissues from Non-Hispanic Black and Non-Hispanic White Women Reveals Correlations between Microbial Dysbiosis and Enhanced Lipid Metabolism Pathways in Triple-Negative Breast Tumors

**DOI:** 10.3390/cancers14174075

**Published:** 2022-08-23

**Authors:** Alana Smith, Xueyuan Cao, Qingqing Gu, Ernestine Kubi Amos-Abanyie, Elizabeth A. Tolley, Gregory Vidal, Beverly Lyn-Cook, Athena Starlard-Davenport

**Affiliations:** 1Department of Genetics, Genomics and Informatics, College of Medicine, University of Tennessee Health Science Center, Memphis, TN 38163, USA; 2Department of Health Promotion and Disease Prevention, College of Nursing, University of Tennessee Health Science Center, Memphis, TN 38163, USA; 3Department of Preventive Medicine, College of Medicine, University of Tennessee Health Science Center, Memphis, TN 38163, USA; 4Division of Hematology/Oncology, College of Medicine, University of Tennessee Health Science Center, Memphis, TN 38163, USA; 5Division of Biochemical Toxicology, FDA/National Center for Toxicological Research, Jefferson, AR 72079, USA

**Keywords:** metabolome, microbiome, triple-negative breast cancer, lipids, breast cancer

## Abstract

**Simple Summary:**

We previously showed that breast tumor tissues from women display an imbalance in abundance and composition of microbiota compared to normal healthy breast tissues. It is unknown whether these differences in breast tumor microbiota may be driven by alterations in microbial metabolites, leading to potentially protective or pathogenic consequences. The aim of our study was to conduct global metabolic profiling on normal and breast tumor tissues to identify differences in metabolite profiles and to determine whether breast microbial dysbiosis may be associated with enrichment of microbial metabolites in triple-negative breast cancer (TNBC) which disproportionately affects women of African ancestry. We observed significant correlations between elevated lipid metabolism pathways and microbial dysbiosis in TNBC tissues from both non-Hispanic black and white women. This is the first study to report an association between breast microbial dysbiosis and alterations in host metabolic pathways in breast tumors, including TNBC, of non-Hispanic black and non-Hispanic white women.

**Abstract:**

Triple-negative breast cancer (TNBC) is an aggressive form of breast cancer that is non-responsive to hormonal therapies and disproportionately impact women of African ancestry. We previously showed that TN breast tumors have a distinct microbial signature that differs from less aggressive breast tumor subtypes and normal breast tissues. However, it is unknown whether these differences in breast tumor microbiota may be driven by alterations in microbial metabolites, leading to potentially protective or pathogenic consequences. The goal of this global metabolomic profiling study was to investigate alterations in microbial metabolism pathways in normal and breast tumor tissues, including TNBC, of non-Hispanic black (NHB) and non-Hispanic white (NHW) women. In this study, we profiled the microbiome (16S rRNA) from breast tumor tissues and analyzed 984 metabolites from a total of 51 NHB and NHW women. Breast tumor tissues were collected from 15 patients with TNBC, 12 patients with less aggressive luminal A-type (Luminal) breast cancer, and 24 healthy controls for comparison using UHPLC-tandem mass spectrometry. Principal component analysis and hierarchical clustering of the global metabolomic profiling data revealed separation between metabolic signatures of normal and breast tumor tissues. Random forest analysis revealed a unique biochemical signature associated with elevated lipid metabolites and lower levels of microbial-derived metabolites important in controlling inflammation and immune responses in breast tumor tissues. Significant relationships between the breast microbiome and the metabolome, particularly lipid metabolism, were observed in TNBC tissues. Further investigations to determine whether alterations in sphingolipid, phospholipid, ceramide, amino acid, and energy metabolism pathways modulate *Fusobacterium* and *Tenericutes* abundance and composition to alter host metabolism in TNBC are necessary to help us understand the risk and underlying mechanisms and to identify potential microbial-based targets.

## 1. Introduction

Triple negative breast cancer (TNBC) is the deadliest form of breast cancer among women in the US [1,2]. TN breast tumors are negative for estrogen receptor (ER), progesterone receptor (PR), and amplification of the human epidermal growth factor receptor 2 (HER2) [3]. TNBC is associated with poor overall survival and there are no targeted therapies to treat TNBC [4,5,6,7]. Non-Hispanic black (NHB) women are approximately three times more likely than non-Hispanic white (NHW) women to develop TNBC [8] or die from the disease [9]. Therefore, studies to investigate mechanisms that contribute to TNBC risk and that are inclusive of underserved populations most impacted by TNBC is an unmet need that would be transformative in further understanding the biology of TNBC. 

Considering that the breast exhibits the presence of a mucosal immune system that protects against pathogens [10], an imbalance in the breast tissue microbiota due to inflammation and alterations in microbial metabolism pathways may give rise to breast cancer development. We previously showed that an imbalance in the composition and abundance of microbiota exists in the breast tissues of women with breast cancer and differs by breast tumor stage and subtype, including triple-negative breast cancer (TNBC), which disproportionately impacts women of African ancestry [11]. Similar studies have also reported differences in microbial composition in breast cancer [12,13,14,15,16]. However, most of these studies were conducted in populations of European ancestry. While the etiology of microbiome-mediated breast cancer remains to be fully understood, it is plausible that alterations in microbial metabolites may influence host metabolic processes, leading to potentially protective or pathogenic consequences. 

Therefore, in the current study, our objective was to conduct global metabolic profiling on normal and breast tumor tissues to identify differences in metabolite profiles and to determine whether breast microbial dysbiosis may be associated with a metabolic signature in triple-negative breast cancer (TNBC), which disproportionately affects women of African ancestry. To our knowledge, this is the first study to show an association between microbial dysbiosis and alterations in microbial metabolites in breast tumor tissues, including TNBC tissues, from both NHB and NHW women. 

## 2. Materials and Methods

### 2.1. Tissue Collection and Processing

We obtained fresh, snap frozen surgical breast tissue specimens from NHB (n = 22) and NHW (n = 29) women, aged 18 years and over. Breast disease status, pathology reports, and other clinical information was obtained following written informed consent from the patient from the Cooperative Human Tissue Network (Birmingham, AL). Breast tissue samples (n = 51) were immediately stored upon receipt and maintained at -80 °C until further processing. TNBC tissues were obtained from 6 NHB and 9 NHW women. An equal number of Luminal breast tumor tissues were obtained from NHB (n = 6) and NHW (n = 6) women. Women without evidence of breast disease, as confirmed by a pathologist, who underwent reduction mammoplasty for macromastia were considered normal (controls). Normal breast tissue specimens were obtained from NHB (n = 11) and NHW (n = 13) women. This study was conducted in accordance with the Declaration of Helsinki, and the protocol was approved by the Institutional Review Board of the University of Tennessee Health Science Center (IRB #16-04717-NHSR).

### 2.2. DNA Extraction and 16s rRNA Gene Sequencing 

DNA extraction and 16S (V4) rRNA gene sequencing was previously performed on breast tissues as we previously described [11]. All analyses were conducted in the R environment. 

### 2.3. Metabolite Analysis and Liquid Chromatography/Mass Spectrometry (LC/MS, LC/MS2)

Ultra-high-performance liquid chromatography (UHPLC)-tandem mass spectrometry and metabolomic profiling analysis was performed using Metabolon as previously described in detail [13]. 

### 2.4. Metabolite Analysis

The bioinformatic analyses of the microbiome data were completed by utilizing the software and platforms including principal component analysis (PCA) as we previously described [11]. 

The metabolite profiling was log transformed and imputed for missing values. Metabolites without variation or greater than 30% duplicated values were excluded from further analyses. To attenuate the high influential data points, a robust linear model was used to assess the differential metabolite levels for each factor. We further used a robust linear model to evaluate the two independent predictors with or without interaction terms. For each test, the Benjamini–Hochberg 95 method was used to control the false discovery rate (FDR). At each q-value cutoff, the list contains q fraction of false positives. The metabolic pathway analysis was conducted on the MetaboAnalyst website (http://www.metaboanalyst.ca, (accessed on 18 July 2022)).

### 2.5. Statistical Analysis

Statistical analysis was performed using GraphPad Prism 9.4 (GraphPad Software, USA) software or the program “R” version 4.1.2 (2021-11-01)—"Bird Hippie". The R Foundation for Statistical Computing Platform: x86_64-w64-mingw32/x64 (64-bit), http://cran.r-project.org/, (accessed on 15 November 2021). Two types of statistical analyses were usually performed: (1) significance tests and (2) classification analyses. (1) For pair-wise comparisons we typically performed Welch’s *t*-tests and/or Wilcoxon’s rank sum tests. Wilcoxon’s sum rank test was used to detect differentially expressed metabolites or phylum in breast cancer. For differential phylum between two groups, we used a *p*-value cutoff of 0.1. For differential metabolites in TNBC vs normal, or Luminal vs normal, we used a q-value cutoff of 0.05, while we used a *p*-value cutoff of 0.05 for TNBC vs. Luminal. For each selected phylum and metabolite, we calculated the Spearman correlation coefficient (28 samples with both data) and plotted as heatmaps. For classification, we used random forest analyses. For other statistical designs, we performed one-way ANOVA analyses for comparisons of the indicated groups. Only when *p* < 0.05 was it considered statistically significant. NS, not significant; *, *p* < 0.05; **, *p* < 0.01; ***, *p* < 0.001. 

## 3. Results

### 3.1. Patient Demographics and Tissue Characteristics

A total of 51 breast tissue samples were analyzed in this study. Table 1 presents the selected patient demographic and breast tissue characteristics. Breast tissue specimens were collected from a total of 24 women free of cancer who underwent breast reduction mammoplasty (normal). A total of 27 breast tumor tissues were collected from women with TNBC (n = 15) or Luminal A (Luminal) breast tumors (n = 12), a less aggressive form of breast cancer that is responsive to hormonal therapy, for comparison. Information on race, menopausal status, and stage of the breast tumors was also collected in this study. Stages III and IV breast tumors were combined due to the low number of stage IV breast tumors. Approximately 57% of patients were premenopausal and 53% of breast tissue specimens were collected from NHW women while 43% were obtained from NHB women. The mean age between women with and without breast cancer was not significantly different (51 ± 1.24 versus 49 ± 3.86 years, respectively; *p*  > 0.05). 

### 3.2. Metabolic Profiling of Normal and Breast Tumor Tissues

To identify metabolic differences between normal and breast tumor tissues, metabolic profiling of breast tissues was acquired by LC-MS. Principal component analysis (PCA) was conducted to observe the differences between normal, Luminal, and TNBC tissues. The PCA score scatter plots are displayed in Figure 1A. We observed a separation tendency between the three groups which suggests differences in metabolite levels between the groups. Specifically, we observed that normal samples clustered significantly differently than Luminal and TNBC samples PC1 (44%) compared with breast tumor tissues (*p* < 0.05) (Figure 1A). Random forests (RF) analysis assessing the differences in all tissue samples stratified by tumor status yielded a predictive accuracy of 82% (data not shown). The biochemical importance plot in Figure 1B highlights the top 30 metabolites in breast tumors (both Luminal and TNBC combined) versus normal tissue, ranked in order of their importance to the classification scheme. The plot features metabolites involved in lipid, amino acid, and energy metabolism pathways. The top metabolites for separation of breast tumor and normal tissue include kynurenine, a tyrosine metabolite linked to inflammation. Other metabolites differentially detected among tissue types included metabolites of the Krebs cycle and oxidative phosphorylation, including alpha-ketoglutarate, various lipid species such as 1-stearoyl-2-oleoyl-GPI, sphingosine, and energy metabolites such as fumarate and N6-succinyladenosine (Figure 1B). To further explore the metabolic changes between normal and breast tumor tissues, candidate metabolites were imported into MetaboAnalyst 3.0 for enrichment pathway analysis. The corresponding metabolic pathways are displayed in Figure 1C, including those for arginine metabolism, alanine, aspartate and glutamate metabolism, the citric acid cycle, butanoate metabolism, sphingolipid metabolism, glutamine and glutamate metabolism, and tryptophan (Trp) metabolism.

We observed significantly elevated levels in the Trp metabolism pathway in breast tumor tissues compared to normal tissue (Figure 2). The Trp metabolite kynurenine (Kyn) is regarded as an inflammatory marker as inflammatory cytokines, indoleamine 2,3-dioxygenase (IDO) and TDO, catalyze the formation of KYN and anthranilate from Trp in breast tissue. As shown in Figure 2, many metabolites in the Trp metabolism pathway were elevated in breast tumor tissues including Trp, N-formylkynurenine, and Kyn. The byproducts of Trp with microbiome metabolic origin or contribution, indoleproprionate (IPA) and indoxyl sulfate, are significantly decreased in breast tumor tissues compared to normal breast tissues. These findings suggest that IPA and indoxyl sulfate levels may serve a protective function in the breast tissue via modulating the levels of microbiota that metabolize Trp metabolites.

### 3.3. Breast Tumor Tissues Manifested Distinguishable Profiles of Metabolites Compared to Normal Breast Tissues

Because we initially wanted to determine whether differences in metabolomic profiles differed by disease state, we conducted metabolomics analyses on normal and breast tumor tissues using univariate analysis. Table 2 shows the top 10 metabolites that were differentially expressed between normal and breast tumor tissues. The leading metabolites that significantly differed among breast tumors compared to normal breast tissues, included primarily lipids, but also included cofactors and vitamins, peptides, amino acids, and carbohydrates. Specifically, pathways involved in phospholipid metabolism, phosphatidylglycerol and glycerolipid metabolism, plasmalogen and pentose metabolism, alanine and aspartate metabolism, riboflavin, carnitine metabolism, and methionine metabolism were significantly different among breast tumor tissues compared to normal breast tissues (*p* < 0.0001) (Table 2). By contrast, levels of the metabolites glycerophosphoserine, glycerophosphoglycerol, 1-(1-enyl-palmitoyl)-GPE, lyxonate, aspartate, 1-palmitoyl-2-oleoyl-GPG, N,N-dimethyl-pro-pro, FAD, deoxycarnitine, and taurine were significantly lower in normal tissues compared to breast tumor tissues (Table 2). Scatter plots further illustrating these significant differences in metabolite levels between normal, Luminal, and TNBC tissues is shown in Figure 3A. 

To determine whether metabolite levels differed by breast tumor subtype, we performed univariate analysis on TNBC and Luminal breast tissues for comparison. Table 3 highlights the major metabolites differently expressed in TNBC vs Luminal breast tumor tissues. The major metabolic sub pathways that were different between the two groups included sphingolipid synthesis, phosphatidylinositol (PI), phosphatidylinositol (PI), TCA cycle, alanine and aspartate metabolism, pyrimidine metabolism, uracil containing hexosylceramides (HCER), histidine metabolism, hexosylceramides (HCER), methionine, cysteine, and SAM and taurine metabolism. Interestingly, phytosphingosine, 1-stearoyl-2-oleoyl-GPI (18:0/18:1)*, N-acetylasparagine, 3-(3-amino-3-carboxypropyl)uridine*, glycosyl ceramide (d18:1/20:0, d16:1/22:0)*, histamine, glycosyl-N-stearoyl-sphingosine (d18:1/18:0), and N-acetyltaurine were each significantly lower in TNBC tissues compared to Luminal breast tissues (*p* < 0.01) (Table 3). Figure 3B further illustrates these significant differences in metabolite levels between Luminal and TNBC tissues. 

We previously showed that breast tumor tissues, including TNBC breast tissues, have a lower abundance of common microbiota compared to normal breast tissues using 16S rRNA sequencing [11]. In this study, we expanded our analysis to determine whether differences in specific microbiota phyla correlated with alterations in lipid metabolites based on breast tumor subtype. We observed significant differences in the pattern of microbiota and lipid metabolites (Figure 4). As shown by the heatmap illustrating the Spearman correlation-based hierarchical clustering analysis, the pattern of microbial phyla and the top 50 significant lipid metabolites revealed differences among the breast tumor subtypes (Figure 4). For example, *Fusobacteria* and *Tenericutes* showed a negative correlation with Lipid-42463 (sphingomyelin (d18:1/14:0, d16:1/16:0)*, Lipid-57432 (ceramide (d18:1/14:0, d16:1/16:0)*, Lipid-57365 (myristoyl dihydrosphingomyelin (d18:0/14:0)*, and Lipid-42446 (1-palmitoyl-2-linoleoyl-GPC (16:0/18:2) in normal tissues (Figure 4A), while *Tenericutes* showed a significant positive correlated with these lipids in TNBC tissues (*p* < 0.05) (Figure 4C), suggesting that a higher abundance of *Tenericutes* may give rise to elevated levels of these metabolites observed in TNBC tissues. Furthermore, there was no correlation between *Crenarchaeota* and lipid metabolites in TNBC tissues (Figure 4C). Circos analysis further revealed correlations among *Tenericutes* and *Fusobacteria* and the top 50 significant metabolites in TNBC and normal breast tissues, respectively (Figure 4B, 4D). On the other hand, *Actinobacteria* and *Crenarchaeota* were positively correlated with Lipid-44621 (1-(1-enyl-oleoyl)-GPE (P-18:1)*, Lipid-52716 (1-(1-enyl-palmitoyl)-2-palmitoyl-GPC (P-16:0/16:0)*), and Lipid-19265 (1-stearoyl-2-oleoyl-GPS (18:0/18:1)) in Luminal breast tissues (Figure 4E,4F). Lastly, we determined whether microbiota phyla were more significantly correlated with certain lipid metabolites in TNBC vs. Luminal breast tissues. Six major microbial phyla were observed in Luminal and TNBC tissues, including phyla *Tenericutes*, *Proteobacteria*, *Firmicutes*, *Bacteroidetes*, *Actinobacteria*, and *Euryarchaeota*. We found that phyla *Firmicutes* and *Bacteroidetes* were positively correlated with Lipid-19503 (stearoyl sphingomyelin (d18:1/18:0) and Lipid-52669 (1-palmitoyl-2-oleoyl-GPI (16:0/18:1)* in TNBC. Additionally, phylum *Bacteroidetes* was positively correlated with Lipid-61700 (2-hydroxyarachidate*) and Lipid-52467 (1-palmitoyl-2-arachidonoyl-GPI (16:0/20:4)* in TNBC (Figure 4G,4H). *Proteobacteria*, the most abundant microbiota phylum in the breast, was also strongly correlated with Lipid-52477 (1-(1-enyl-palmitoyl)-2-oleoyl-GPE (P-16:0/18:1)* in TNBC tissues. Conversely, several phyla, including *Firmicutes* and *Tenericutes*, were negatively correlated with many of these and several other lipid metabolites in Luminal breast tissues (Figure 4G,H). These findings suggest that phyla *Firmicutes* and *Tenericutes* may promote an imbalance in the levels of certain lipid metabolites, such as sphingomyelin (d18:1/14:0, d16:1/16:0), stearoyl sphingomyelin (d18:1/18:0), 1-palmitoyl-2-oleoyl-GPI (16:0/18:1), and ceramide (d18:1/14:0, d16:1/16:0), leading to TNBC development. 

## 4. Discussion 

In the present study, we sought to identify differences in metabolite profiles between normal and breast tumor tissues and to determine whether breast microbial dysbiosis may be associated with enrichment of microbial metabolites in TNBC tissues. Our initial analysis revealed significantly elevated levels of metabolites involved in Trp metabolism in breast tumor tissues, including KYN, compared to normal breast tissues. Interestingly, KYN is regarded as an inflammatory marker, which is a common characteristic in cancer [17,18]. While KYN can function to modulate the inflammatory immune response in normal tissues, activity of this pathway in tumor cells tends to be overexpressed and can lead to the depletion of tryptophan in the microenvironment, thereby suppressing T-cell mediated immune response [19]. In our study, we observed that many of the metabolites belonging to the Trp metabolism pathway were significantly elevated among breast tumor tissues when compared to normal breast tissues. The metabolites in this pathway were also significantly altered between the different tumor expression profiles, particular among TNBC tissues (data not shown). In a study by Kanaan et al., 3-fold greater Trp levels and a 7-fold and 8-fold increase in its metabolic intermediates in NAD+ biosynthesis, KYN and quinolinate, respectively [20] were reported, which is in agreement with our findings (data not shown). Furthermore, a positive correlation between the increased expression of enzymes involved in KYN metabolism, including kynurenine 3-monooxygenase (KMO), has been observed to play a role in TNBC metastasis and recurrence [21] and malignant phenotype in patients with TNBC [22]. Of interest, overexpression of KMO has been suggested to contribute to the malignant phenotype of TNBC cells, particularly cancer stem cell (CSC) properties, and 3) KMO-induced β-catenin stabilization, which also leads to increased expression of pluripotent genes such as Nanog, Oct4, and Sox2 [23].

In line with increased metabolic activity of tumor tissue, within the dataset there were significant alterations in several metabolites linked to energy metabolism. Glucose was not among the biochemicals that were significantly altered; however, metabolites derived from glucose via the glycolytic pathway (dihydroxyacetone phosphate, pyruvate), pentose phosphate pathway (lyxonate), and hexosamine pathways (UDP-glucuronate, UDP-N-acetylglucosamine/galactosamine) were among those that were most significantly elevated. UDP-N-acetylglycosamine (UDP-GlcNAc) and UDP-glucuronate, both nucleotide sugar with pivotal functions as a key substrate for the synthesis of glycoconjugates and polysaccharides, have been linked to breast cancer through its dependence on glycolysis intermediate availability [24]. These increases are consistent with increased rates of glucose utilization. Importantly, changes of this nature may facilitate anabolic growth in tumor cells. Our findings agree with those of Tayyari et al., who observed profound metabolic alterations characterized by decreased mitochondrial respiration and increased glycolysis in TNBC of African American women compared to Caucasian women [25]. Metabolites belonging to the catabolism of branched chain amino acids butyrylcarnitine and propionylcarnitine were also significantly elevated in tumor tissue. Additionally, longer acylcarnitine species, such as palmitoylcarnitine (C16) and stearoylcarnitine (C18), associated with lipid beta oxidation, were also significantly higher in tumor tissue. 

Increases in collagen metabolites were also observed and could be indicative of the breakdown of extracellular matrix and tissue remodeling. Many metabolites found almost exclusively within collagen were significantly increased in tumor tissue when compared to healthy breast tissue. Trans-4-hydroxyproline, pro-hydroxyproline, and N,N-dimethyl-pro-pro were all significantly elevated. Interestingly, a link between collagen levels and breast cancer metastasis has been established in several published reports [26,27,28]. Collagen degradation products, in fact, have been explored as potential breast cancer biomarkers in other studies [29]. Significant increases were also observed in several of the phospholipid synthetic intermediates, including choline phosphate, cytidine 5′-diphosphocholine, and cytidine-5′-diphosphoethanolamine. Intermediates involved in sphingosine synthesis (sphinganine) and triacylglycerol metabolism were also significantly elevated among tumor samples when compared to normal control tissue. Metabolites generated during phospholipid and sphingolipid turnover (e.g., lysolipids, plasmalogens, sphingosine) were also elevated. These changes could also be indicative of tissue remodeling and increased lipid synthesis. 

Metabolites of the Krebs cycle and oxidative phosphorylation were also significantly altered among tumor tissue when compared to healthy breast tissue, and some metabolites were differentially regulated between tumor expression patterns. Alpha-ketoglutarate was significantly elevated in TNBC tumors when compared to Luminal tumors. In addition, tumor tissue exhibited significant differences with regard their nucleotide and nicotinamide profiles. Purine nucleotides, such as AMP, cAMP, and GMP, were significantly elevated (data not shown), while FAD had greater than 3-fold change in TNBC tissue compared to normal breast tissue. The increase in these metabolites likely correlates with increases in their synthesis and utilization likely to promote tumor growth and proliferation [30].

Considering that aberrant lipid accumulation and marked changes in cellular lipid profiles are related to breast cancer metabolism and disease progression [31], it is not surprising that the metabolites most significantly altered in TNBC tissues were involved in lipid metabolism, including phytosphingosine, 1-palmitoyl-2-arachidonoyl-GPI (16:0/20:4)*, and 1-stearoyl-2-oleoyl-GPI (18:0/18:1)*. These changes could be indicative of tissue remodeling and alterations in lipid synthesis and catabolism. Additionally, longer acylcarnitine species, such as palmitoylcarnitine (C16) and stearoylcarnitine (C18), associated with lipid beta oxidation, were also significantly higher in breast tumor tissue. Our findings agree with those by Zhang et al., who reported that butyrylcarnitine (C4) levels were associated with increased odds of breast cancer (OR 1.12; 95% CI 1.02–1.23) in a retrospective case-control study of 991 female breast cancer cases and 991 female controls [32]. These increases are consistent with increased rates of glucose utilization [33]. Importantly, changes of this nature may facilitate anabolic growth in breast tumor cells [34]. 

In addition to these findings, our study demonstrated for the first time a significant correlation between alterations in lipid metabolites and abundance of certain microbiota phyla in TNBC and Luminal breast tissues of NHB and NHW women. Specifically, we observed a significant negative correlation between *Fusobacteria* and *Tenericutes* and the lipids sphingomyelin (d18:1/14:0, d16:1/16:0)*, ceramide (d18:1/14:0, d16:1/16:0)*, and 1-palmitoyl-2-linoleoyl-GPC (16:0/18:2) in normal tissues (Figure 4A), while Tenericutes showed a significant positive correlation with these lipids in TNBC tissues (*p* < 0.05) (Figure 4C), suggesting that a higher abundance of *Tenericutes* may give rise to elevated levels of these metabolites observed in TNBC tissues. *Tenericutes*, a phylum of Gram-negative bacteria, has been associated with diet-induced maternal obesity, a common characteristic of TNBC, in a murine model [35]. Interestingly, we observed significantly elevated levels of these lipids, particularly sphingomyelin and 1 and 2-palmitoleoyl-GPC, in NHB women compared to NHW women (data not shown), which could explain a portion of the racial disparities in TNBC risk and outcomes. Furthermore, a higher abundance of *Tenericutes* was reported in advanced stages of breast cancer [14]. *Fusobacterium* contains a lipopolysaccharide (LPS) that functions as a bacterial toxin by activating immune responses which has been observed in autoimmune disorders, obesity, depression, and cellular senescence [36,37,38,39]. Bacterial LPS has been shown to stimulate ceramide-activated protein kinase activity via hydrolysis of sphingomyelin in human leukemia (HL–60) cells [40]. Thus, it is plausible that elevated levels of sphingomyelin may contribute to the activity of LPS in *Fusobacterium*. Interestingly, strains of *Fusobacterium* have been associated with several human diseases, including estrogen receptor positive breast cancer, colon cancer, periodontal diseases, and topical skin ulcers [15,41,42,43]. Hibberd et al., reported a reduction in *Fusobacterium* in feces of colorectal cancer patients that received probiotics compared to those who did not receive probiotics [44]. These findings suggest that intake of probiotics may be a potential therapeutic approach to manipulate *Fusobacterium* microbiota thereby reducing LPS and the risk of TNBC development.

We recognize that a limitation of our study is the small number of breast tissue samples analyzed. However, we were able to validate findings from similar metabolomic studies to identify signatures associated with breast cancer, including TNBC [20,25,45,46,47,48,49]. Furthermore, this study is the first to show a significant correlation between alterations in lipid metabolites and the abundance of certain microbiota phyla in TNBC and Luminal breast tissues of NHB and NHW women. Our study expands on our previous published findings on the breast tissue microbiome in which we further identified distinct microbial signatures that differ by race, stage, and breast tumor subtype [11]. Importantly, the present study consists of breast tissue specimens from populations of African and European ancestry and, therefore, is representative of a diverse and inclusive population. 

## 5. Conclusions

In conclusion, we report for the first time significant correlations between the breast tissue microbiome and the lipid metabolome in TNBC and luminal breast tissues compared to normal breast tissues of NHB and NHW women. We observed a significant correlation between alterations in lipid metabolites and the abundance of *Tenericutes* and *Fusobacterium* in TNBC tissues of NHB and NHW women. Further investigations to determine whether alterations in sphingolipid, phospholipid, ceramide, and energy metabolism pathways modulate *Tenericutes* and *Fusobacterium* abundance and composition to alter host metabolism in TNBC are necessary to help us understand the risk and underlying mechanisms and to identify potential microbial-based targets.

## Figures and Tables

**Figure 1 cancers-14-04075-f001:**
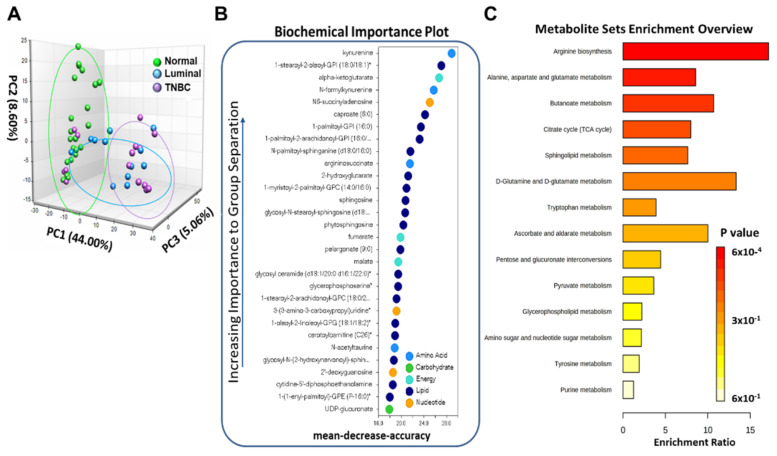
Metabolome analysis of normal and breast tumor tissues. (**A**) Principal component analysis of the metabolome analysis for normal (green), Luminal (blue), and TNBC (purple) tissues. (**B**) Random forest analysis illustrates the top 30 metabolites with the highest discriminatory power between normal, Luminal and TNBC groups. (**C**) Metabolism pathways significantly enriched in breast tumors compared to normal breast tissues.

**Figure 2 cancers-14-04075-f002:**
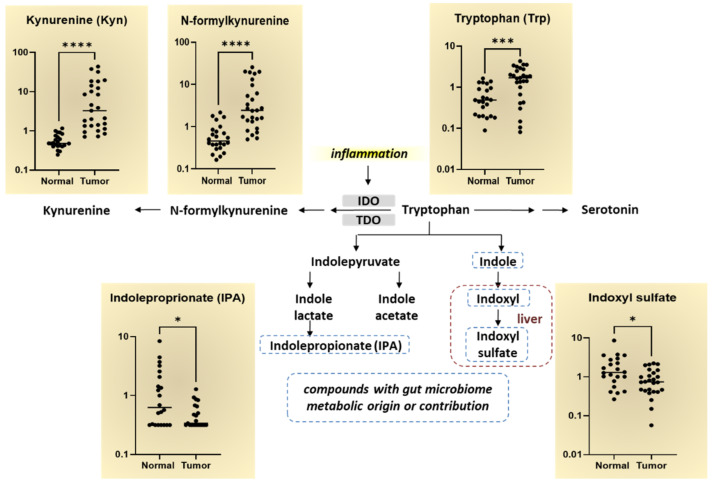
Tryptophan metabolism pathway alterations in breast tumor tissues compared to normal breast tissues. Log-scale differences in the major Trp metabolites in normal and breast tumor tissues are shown in the scatter plot graphs. The Trp metabolism pathway begins with the rate-limiting inflammatory cytokines, IDO and TDO, which involves cleaving Trp by indoleamine 2,3-dioxygenase (IDO1/IDO2) or tryptophan 2,3-dioxygenase 2 (TDO2) to form N-formylkynurenine. N-formylkynurenine is further metabolized by enzymes to kynurenine. The byproducts of Trp with microbiome metabolic origin or contribution, indoleproprionate (IPA) and indoxyl sulfate, are indicated by the dotted line. * *p* < 0.05, *** *p* < 0.0005, **** *p* < 0.0001 is considered statistically significant.

**Figure 3 cancers-14-04075-f003:**
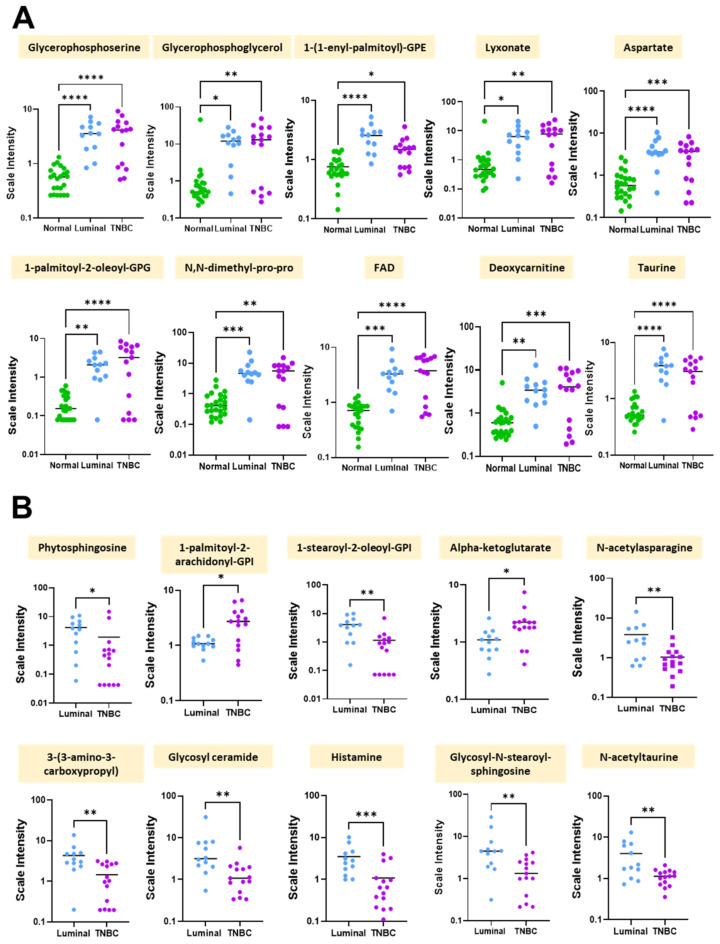
Significant differences in metabolite levels between normal and breast tumor tissue groups. Shown are scatter plots of the mean +/− standard error measurement. (**A**) Individual differences in scale intensities of the most significantly altered metabolites between normal (green), Luminal (blue), and TNBC (purple) breast tissues. (**B**) Individual differences in scale intensities of the most significantly altered metabolites between Luminal (blue) and TNBC (purple) breast tissues. * *p* < 0.05, ** *p* < 0.005, *** *p* < 0.0005, **** *p* < 0.0001 is considered statistically significant.

**Figure 4 cancers-14-04075-f004:**
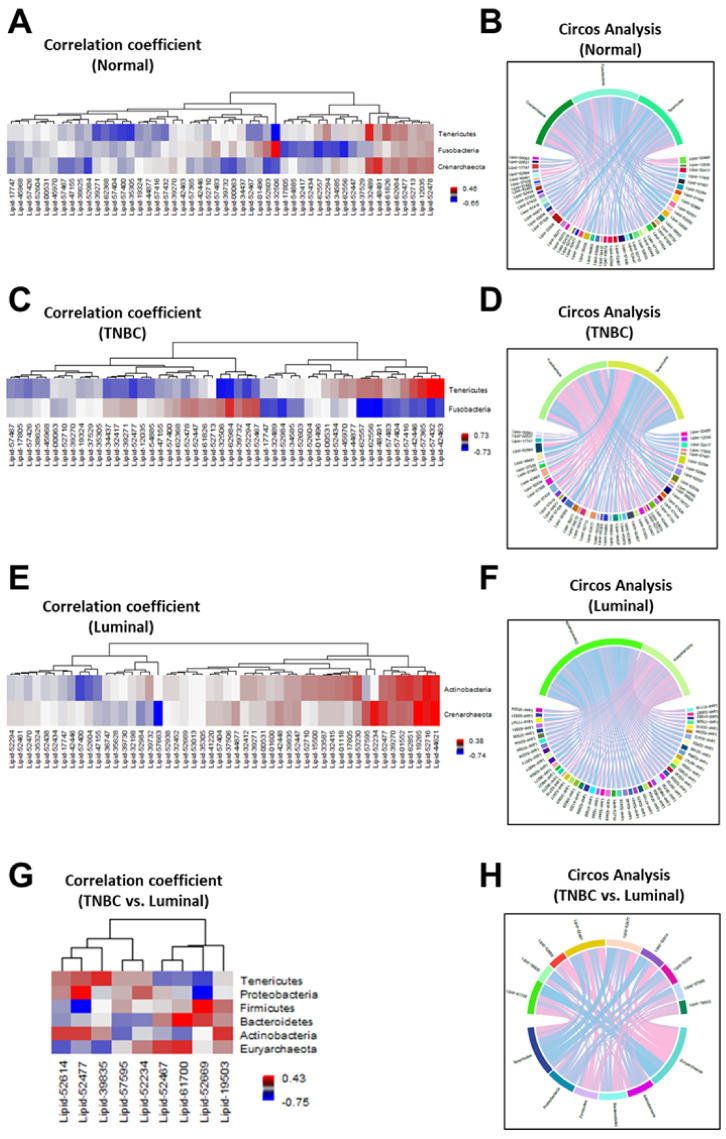
Correlation between microbiota phyla and lipid metabolites in normal and breast tumor tissues. Spearman correlation-based hierarchical clustering analysis and Circos diagrams revealed diversity differences in microbiota phyla and lipid metabolites in (**A,B**) normal, (**C,D**) TNBC (**E,F**) Luminal, and (**G,H**) TNBC vs. Luminal breast tumor tissues. Blue color represents a negative correlation between the microbiota and lipid shown and red color indicates a positive correlation between the microbiota and lipid shown.

**Table 1 cancers-14-04075-t001:** Patient breast tissue characteristics.

Variable	Total Patients (n = 51)	Normal (n = 24)	Tumor (n = 27)
Mean age, years			
Average (range)	50 (21–80)	49 (21–75)	51 (25–80)
Ethnicity			
NHB	22	11	11
NHW	29	13	16
Menopausal status			
Premenopausal	29	14	15
Postmenopausal	22	10	12
Stage			
1	10	NA	10
2	10	NA	10
3/4	7	NA	7
Tumor subtype			
Luminal	12	NA	12
TNBC	15	NA	15

Abbreviations: NHB: non-Hispanic blacks; NHW: non-Hispanic whites; TNBC: triple-negative breast cancer; NA: not applicable; SD: standard deviation.

**Table 2 cancers-14-04075-t002:** Significantly different metabolites between normal and breast tumor tissues.

Biochemical	Super Pathway	SUB Pathway	Mean ± SD	*p*-Value
Glycerophosphoserine	Lipid	Phospholipid metabolism	3.284 ± 0.278	0
Glycerophosphoglycerol	Lipid	Glycerolipid metabolism	11.851 ± 1.061	0
1-(1-enyl-palmitoyl)-GPE (P-16:0)	Lipid	Lysoplasmalogen	2.885 ± 0.260	0
Lyxonate	Carbohydrate	Pentose metabolism	6.394 ± 0.559	0
Aspartate	Amino acid	Alanine and aspartate metabolism	2.884 ± 0.264	0
1-palmitoyl-2-oleoyl-GPG (16:0/18:1)	Lipid	Phosphatidylglycerol (PG)	2.077 ± 0.187	0
N,N-dimethyl-pro-pro	Peptide	Modified peptides	4.193 ± 0.400	0
FAD	Cofactors and vitamins	Riboflavin metabolism	2.761 ± 0.266	0
Deoxycarnitine	Lipid	Carnitine metabolism	3.110 ± 0.297	0
Taurine	Amino acid	Methionine, cysteine, SAM, and taurine metabolism	2.714 ± 0.266	0

**Table 3 cancers-14-04075-t003:** Significantly different metabolites between TNBC and Luminal breast tissues.

Biochemical	Super Pathway	Sub Pathway	Mean ± SD	*p*-Value
Phytosphingosine	Lipid	Sphingolipid synthesis	−3.214 ± 0.561	0.000015
1-palmitoyl-2-arachidonoyl-GPI (16:0/20:4)	Lipid	Phosphatidylinositol (PI)	1.543 ± 0.318	0.000055
1-stearoyl-2-oleoyl-GPI (18:0/18:1)	Lipid	Phosphatidylinositol (PI)	−2.862 ± 0.589	0.000074
Alpha-ketoglutarate	Energy	TCA cycle	0.939 ± 0.217	0.000197
N-acetylasparagine	Amino Acid	Alanine and aspartate metabolism	−1.700 ± 0.423	0.000828
3-(3-amino-3-carboxypropyl) uridine	Nucleotide	Pyrimidine metabolism, uracil containing	−2.258 ± 0.602	0.001013
Glycosyl ceramide (d18:1/20:0, d16:1/22:0)	Lipid	Hexosylceramides (HCER)	−2.437 ± 0.707	0.002439
Histamine	Amino Acid	Histidine metabolism	−1.867 ± 0.577	0.003713
Glycosyl-N-stearoyl-sphingosine (d18:1/18:0)	Lipid	Hexosylceramides (HCER)	−2.497 ± 0.795	0.004926
N-acetyltaurine	Amino Acid	Methionine, cysteine, SAM, and taurine metabolism	−1.262 ± 0.425	0.009224

## Data Availability

The data presented in this study are available upon request from the corresponding author.

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
