# Peer review of "Characterization of the Metabolome of Breast Tissues from Non-Hispanic Black and Non-Hispanic White Women Reveals Correlations between Microbial Dysbiosis and Enhanced Lipid Metabolism Pathways in Triple-Negative Breast Tumors"

_cancers, 2022, doi:10.3390/cancers14174075_

Round 1
Reviewer 1 Report
The authors discuss a study designed to evaluate metabolomic differences between normal vs breast tumor tissues and between luminal A vs triple negative breast cancer tissues among non-Hispanic Black and White women. The studies performed are of sufficient novelty to warrant publication once after addressing some important points as outlined below.
On line 189, what should come after the word and? There is simply a period after that.
The authors report that differences in the microbiota phyla such as Firmicutes and Tenericutes lead to imbalances in lipid metabolites to promote TNBC development rather than luminal breast cancer development. Obesity is particularly prevalent among AA patients with TNBC. Given this, were there significant differences observed in the metabolomic profile based on race/ancestry? The authors should discuss whether significant differences were found and if not speculate as to why despite findings to suggest that TNBC among AA patients tends to be more aggressive.
Did the authors notice a higher percentage of tumors among AA patients derived from the TNBC subtype as compared to the luminal A subtype or was the sample size too small to draw such conclusions?
Author Response
We thank reviewer 1 for his/her insightful comments and suggestions. We have included a detailed response to the reviewers concerns independently as addressed below:
The authors discuss a study designed to evaluate metabolomic differences between normal vs breast tumor tissues and between luminal A vs triple negative breast cancer tissues among non-Hispanic Black and White women. The studies performed are of sufficient novelty to warrant publication once after addressing some important points as outlined below.
On line 189, what should come after the word and? There is simply a period after that.
COMMENT: This was an error and has been deleted.
The authors report that differences in the microbiota phyla such as Firmicutes and Tenericutes lead to imbalances in lipid metabolites to promote TNBC development rather than luminal breast cancer development. Obesity is particularly prevalent among AA patients with TNBC. Given this, were there significant differences observed in the metabolomic profile based on race/ancestry? The authors should discuss whether significant differences were found and if not speculate as to why despite findings to suggest that TNBC among AA patients tends to be more aggressive.
COMMENT: This is a great and insightful question raised by the reviewer. We did observe significant differences by race in metabolomics profiles, particularly in regards to lipids. However, due to the small sample size, we did not show this data. We are planning larger studies with more breast tissue specimens to address this specific question.
We provided a statement in the Discussion section that states “Interestingly, we observed significantly elevated levels of these lipids, particularly sphin-gomyelin and 1 and 2-palmitoleoyl-GPC in NHB women compared to NHW white (data not shown), which could explain a portion of the racial disparities in TNBC risk and out-comes.”
Did the authors notice a higher percentage of tumors among AA patients derived from the TNBC subtype as compared to the luminal A subtype or was the sample size too small to draw such conclusions?
COMMENT: We thank the reviewer for acknowledging this important point. There was no significant difference in the percentage of tumors by race. We have added a statement in the methods section under Tissue collection and processing section that “TNBC tissues were obtained from 6 NHB and 9 NHW women. An equal number of luminal breast tumor tissues were obtained from NHB (n=6) and NHW (n=6) women. Normal breast tissue specimens were obtained from NHB (n=11) and NHW (n=13) women”.
Reviewer 2 Report
In the manuscript entitled “Characterization of the metabolome of breast tissues from non-Hispanic black and non-Hispanic white women reveals correlations between microbial dysbiosis and enhanced lipid metabolism pathways in triple negative breast tumors” the authors corelated the dysbiotic microbiome of the breast cancer tumors with the metabolic pathways.
The authors addressed the differential metabolic pathways and tried to establish link with the altered microbiome in the breast tumors specially TNBC. The study is very well designed and executed. The sample size although seems too small provided the big conclusion, but the sample selection and the rational of the sample selection is very well explained. There are few areas that to my understanding can be improved.
1. The breast microbiome diversity depends on a lot of different factors including food habits, habitat, use of different medications and other chemicals. So criteria for selection of control sample is very crucial. I think the study would have been much better if the authors would have selected also matched controls (non-cancerous tissues surrounding the tumor). This would have given a clear picture of the effect of the dysbiotic microbiome on the detected metabolites.
2. The sample size seems to be very small compared to the importance and conclusion of the study.
3. I am confused about what the authors wanted to corelate between the gut and the breast cancer microbiome? They may explain it clearly for the readers to understand in a better way.
4. In the manuscript the authors in some places regarded the LuminalA and the TNBC as one group (Tumor) and the normal tissues as another group. In some sections authors also divided the tumor group between Luminal A and TNBC and compared with the non tumor samples. Is there any specific reason for that. If they wanted to show the more aggressive TNBC has a distinct metabolic composition compared to the Luminal A, then they should do that in all the figures. If the authors are more interested in showing the difference between breast tumor and normal then they should be consistent.
The authors should explain properly why they preferred to analyze in two different groups in two sections of the manuscript.
5. In figure 3A for Asperate the significance is *** or **** when the values in TNBC seems to be very scattered. While in cane of Figure 3B (1-palmitoyl-2-arachidonyl-GPI) the point seems to be more concentrated, but the significance is only *. The authors should recheck the statistical study they performed and see through all the analysis for any possible mistake.
6. There are some more previous studies that dealt with TNBC and metabolism. Those should be cited properly in the manuscript.
Author Response
We thank reviewer 2 for his/her insightful comments and suggestions. We have included a detailed response to the reviewers concerns independently as addressed below:
- The breast microbiome diversity depends on a lot of different factors including food habits, habitat, use of different medications and other chemicals. So criteria for selection of control sample is very crucial. I think the study would have been much better if the authors would have selected also matched controls (non-cancerous tissues surrounding the tumor). This would have given a clear picture of the effect of the dysbiotic microbiome on the detected metabolites.
COMMENTS: We agree with the reviewer that matched controls would have provided more evidence of the effect of breast microbial dysbiosis and the detected metabolites; however, we did not have access to sufficient number of match control samples for these studies. Nonetheless, our previous published findings described by Smith et al (PMID: 31420578) showed that the microbial profiles do not significantly differ between normal and normal adjacent (normal pair) breast tissue specimens. Therefore, we felt it was sufficient to include health normal controls, without prior evidence of breast cancer, as a more fitting control for these analyses.
- The sample size seems to be very small compared to the importance and conclusion of the study.
COMMENTS: We agree with the reviewer that the sample size is small; however, we were able to replicate findings from studies that had a larger sample size. We discuss the limitations of the study in the Discussion section and suggests future studies to address the small sample size.
- I am confused about what the authors wanted to corelate between the gut and the breast cancer microbiome? They may explain it clearly for the readers to understand in a better way.
COMMENTS: We apologize for the confusion. Many of the bacteria that we observed to be dysregulated in the breast are also prominent in the gut. To alleviate the confusion, we have removed the wording regarding to gut and breast cancer throughout the text.
- In the manuscript the authors in some places regarded the LuminalA and the TNBC as one group (Tumor) and the normal tissues as another group. In some sections authors also divided the tumor group between Luminal A and TNBC and compared with the non tumor samples. Is there any specific reason for that. If they wanted to show the more aggressive TNBC has a distinct metabolic composition compared to the Luminal A, then they should do that in all the figures. If the authors are more interested in showing the difference between breast tumor and normal then they should be consistent.
The authors should explain properly why they preferred to analyze in two different groups in two sections of the manuscript.
COMMENTS: We thank the reviewer for their suggestions. We initially wanted to determine which metabolites were most significantly different by disease state (Table 2), therefore we compared the metabolomic profiles between normal and tumor tissues (both luminal and TNBC). Figure 3 further provides scatter plots between normal, luminal and TNBC tissues of those significant metabolites listed in Table 2. We have also provided the statement in the results section to make this clear to the reader: “Because we initially wanted to determine whether differences in metabolomic profiles differed by disease state, we conducted metabolomics analysis on normal and breast tumor tissues…”
- In figure 3A for Asperate the significance is *** or **** when the values in TNBC seems to be very scattered. While in cane of Figure 3B (1-palmitoyl-2-arachidonyl-GPI) the point seems to be more concentrated, but the significance is only *. The authors should recheck the statistical study they performed and see through all the analysis for any possible mistake.
COMMENTS: Thank you for your suggestions. The scatter plot for aspartate in Figure 3A compares normal vs luminal, and normal vs. tnbc. The significance is much higher for these comparisons since the differences observed are between normal and the tumor group. Whereas in Figure 3B, the scatter plots are comparing luminal vs tnbc. Nonetheless, we confirmed the statistical analysis is correct.
We confirm that for statistical analysis for aspartate using multiple comparison analysis when comparing normal vs luminal the p-value is <0.0001 and normal vs tnbc is 0.00008
We confirm that for statistical analysis for 1-palmitoyl-2-arachidonyl-GPI using two-tailed t-test when comparing tnbc vs luminal the p-value is 0.0321
- There are some more previous studies that dealt with TNBC and metabolism. Those should be cited properly in the manuscript.
COMMENTS: We thank the reviewer for his/her comments. We agree with the reviewer that other studies should be cited properly throughout the manuscript in regards to TNBC and metabolism. However, when inputting these terms into Pubmed, over 9,000 citations are found. Nonetheless, we have updated the manuscript to include additional metabolomic studies that have been conducted in TNBC tissues, including in African American women. Specifically, we have cited PMID: 33181091, 35105939, 31911556, 29545929, 25422359